# ECOD domain classification of 48 whole proteomes from AlphaFold Structure Database using DPAM2

R. Dustin Schaeffer[1]*, Jing Zhang[1,2], Kirill E. Medvedev[1], Lisa N. Kinch[3,4], Qian Cong[1,2], Nick V. Grishin[1,5]

1 Department of Biophysics, University of Texas Southwestern Medical Center, Dallas, Texas, United States of America, 2 Eugene McDermott Center for Human Growth and Development, University of Texas Southwestern Medical Center, Dallas, Texas, United States of America, 3 Department of Molecular Biology, University of Texas Southwestern Medical Center, Dallas, Texas, United States of America, 4 Howard Hughes Medical Institute, University of Texas Southwestern Medical Center, Dallas, Texas, United States of America, 5 Department of Biochemistry, University of Texas Southwestern Medical Center, Dallas, Texas, United States of America

* Richard.Schaeffer@UTSouthwestern.edu

**Data Availability Statement:** Domain data (including marginal domains) including homology assignments and domain ranges are deposited in Zenodo and available at the ECOD website as a

## Abstract

Protein structure prediction has now been deployed widely across several different large protein sets. Large-scale domain annotation of these predictions can aid in the development of biological insights. Using our Evolutionary Classification of Protein Domains (ECOD) from experimental structures as a basis for classification, we describe the detection and cataloging of domains from 48 whole proteomes deposited in the AlphaFold Database. On average, we can provide positive classification (either of domains or other identifiable non-domain regions) for 90% of residues in all proteomes. We classified 746,349 domains from 536,808 proteins comprised of over 226,424,000 amino acid residues. We examine the varying populations of homologous groups in both eukaryotes and bacteria. In addition to containing a higher fraction of disordered regions and unassigned domains, eukaryotes show a higher proportion of repeated proteins, both globular and small repeats. We enumerate those highly populated domains that are shared in both eukaryotes and bacteria, such as the Rossmann domains, TIM barrels, and P-loop domains. Additionally, we compare the sampling of homologous groups from this whole proteome set against our stable ECOD reference and discuss groups that have been enriched by structure predictions. Finally, we discuss the implication of these results for protein target selection for future classification strategies for very large protein sets.

## Author summary

Proteins can contain one or more domains, regions that are evolutionary independent and convey fiction and function. Here we present our classification of proteins within 48 proteomes provided by the AlphaFold Structural Database. These proteomes span multiple model organisms used in research as a common ground for studying biological

special collection. (https://zenodo.org/doi/10.5281/zenodo.8384982) Analysis data and figure generation codes are collected within the Supporting Information.

**Funding:** This research is funded by NIH GM147367 to RDS and NIH GM127390 to NVG. QC is a Southwestern Medical Foundation-endowed scholar. JZ is supported by a training grant RP210041 from the Cancer Prevention and Research Institute of Texas. This research is funded by NSF 2224128 (DBI) to NVG. This research is also supported by grant I-2095-20220331 to QC and I-1505 to NVG from the Welch Foundation. NSF: https://www.nsf.gov/div/index.jsp?div=DBI NIH: https://www.nigms.nih.gov/ CPRIT: https://www.cprit.state.tx.us/ Welch Foundation: https://welch1.org/ None of these funders played any role in study design, data collection or analysis, the decision to publish, or in manuscript preparation.

**Competing interests:** The authors have declared that no competing interests exist.

principles as well as organisms involved in prevalent human infectious diseases. We classify these domains by our Domain Parser for AlphaFold Models (DPAM), which was previously tested on the human proteome. We find that eukaryotic and bacterial proteomes can be classified to different degrees, with significantly more disordered and low-confident regions in eukaryotic proteins.

## Introduction

Protein domains are independent evolutionary units that convey function and fitness either singly or in concert. The study of domains and their homology is a powerful tool for studying protein function [1–3]. Homologous domains can share function, and the propagation of functional annotation from experimentally characterized proteins and their domains to their homologous yet hypothetical or uncharacterized domains can lead to biological insights [4–7]. Protein domain classifications determine and organize these homologous domains [8] and either 1) sequence classifications such as Pfam [9], CDD [10], or SUPERFAMILY [11] that partition protein sequence into domains and derive their taxonomy principally by sequence similarity measures or 2) structure classifications such as SCOP [12,13], CATH [14], or ECOD [15] that use structural similarity to determine more distant homology (at the cost of access to fewer proteins). The advent of highly accurate structure prediction software may upset this traditional division. The outstanding performance of DeepMind's AlphaFold2 at CASP14[16–18] led to the subsequent development of software such as RoseTTAFold [19], ESMFold [20], and I-TASSER-MTD [21] as well as the widespread prediction of large protein sets resulting in resources such as the AlphaFold Structure database(AFDB)[22] and ESM-Atlas [23].

AlphaFold is a deep-learning-based structure prediction method trained on data generated by structural biologists and stored by the Protein Data Bank (PDB)[24]. Since its release in 2020, it has been used to generate and make available numerous data sets through the AlphaFold Structural Database hosted by the EBI, including 48 whole proteomes, numerous curated proteins within the SWISS-PROT dataset, and 200M predictions representing the known protein universe [22]. Two major measures of confidence are made available with each prediction: 1) Per-residue predicted local distance difference test (plDDT) measures, which can be used to measure prediction confidence and 2) predicted aligned errors (PAE) which can be used to measure relative pairwise confidence between two residues (or regions) within a prediction. These measures can identify disordered and low-confidence regions with proteins that may only be structured in a multimeric context, or that are biologically disordered. These measures can be used as an effective screening tool for models or regions of models towards structurally ordered or globular regions [25]. Extensive analysis of these measures in the predictions of human proteins revealed that ~75% of the human proteome was covered by experimental structures, predicted models, or homology models of existing experimental structures, in addition to 12% of the proteome that is judged to be intrinsically disordered. Only a small fraction (2%) of the proteome was judged to be *bona fide* domains with unknown structure or very distant homology [26]. These proportions correspond with our observation of ECOD domains in the predicted structure of the human proteome [27]. We use them directly in DPAM for partition and assignment, as well as in our analysis and screening of the resulting domains and regions.

For nearly a decade, we have classified domains into our Evolutionary Classification of Protein Domains (ECOD) from experimental structures using a combination of structural and sequence aligners along with expert curation on selected difficult cases [15,28]. As opposed to

other structural domain classifications, ECOD's hierarchy is structured to prioritize homology over topology. Although the principal goal of ECOD has been the classification of proteins with experimental structures, we have recently demonstrated the utility of ECOD for domain classification in both eukaryotic [27] and bacterial (*Vibrio parahaemolyticus RIMD)* proteomes. In both cases, ECOD, combined with a recently developed structural domain parser (DPAM) for AlphaFold models [29,30], classified the available residues into either domains or low-confidence and disordered regions. The resulting domains from these proteomes suggested biological insights for their respective organisms, and they were made available on the ECOD website. The flood of protein predictions after the release of AlphaFold, and the observed domain content relative to unstructured or low-confidence regions, prompted us to examine the behavior of our domain classification more closely on the broadest selection of whole proteomes. Among the available models, there is a breadth of structure quality, arising from multiple potential causes: 1) lack of structural context for multimeric complexes 2) biological disorder within proteins 3) conformation changes, 4) genome annotation, and 5) error in structure prediction. We suggest that domain classification from these structures, and positively identifying different types of regions within them, is necessary to use these predictions most effectively. Moreover, closely associating domains from predictions with experimental data, where possible, is one way to mitigate possible errors from the structure prediction algorithm.

The classification of proteins into domains is one potential strategy for grappling with the size and quality of the data offered by large structure prediction efforts. Previous efforts using a prior version of the AF2 inference model (v1) and a smaller selection of 21 proteomes have found that in a large percentage (92%) of models, these proteomes can be partitioned into domains and assigned to the existing CATH hierarchy [31]. This work also featured manual assignment of difficult-to-assign human domains, as well as classification of domains into closely related sequence families. The division of predicted models into easily classified domains, domains with more distant or difficult-to-recognize homology, or regions that are unstructured or have no domains whatsoever will continue to be a useful strategy for grappling with these large structural datasets.

Here we classify the domains from 48 whole proteomes available in the AFDB. These proteomes include a wide selection of model organisms such as zebrafish, fruit flies, and *E. coli*. They also include numerous organisms that cause diseases such as malaria, leprosy, and bacterial pneumonia. We examine the varying population of homologous groups among predictions from whole proteomes and the experimentally characterized protein universe. We also examine the differing classification between eukaryotes and bacteria in these predicted examples. The widespread access to proteins that were previously difficult or unamenable to structural classification dramatically diminishes the gap between structural and sequence classification. Finally, we illustrate those homologous groups that are enriched by domains and use well-known sequence domain annotations to estimate the degree to which structure prediction of proteins and classification of their domains has expanded our knowledge of previously unknown domains.

## Results and discussion

### Domain classification of 48 whole proteomes using DPAM

Domain classification can reveal functional information by identifying homologous relationships between domains. By classifying whole proteomes, we can determine the prevalence of given homologous domain groups within different organisms as well as determine those domains previously unannotated by sequence methods. We classified domains from 48 whole

proteomes from AFDB [32] using the DPAM v2 (Domain Parser for AlphaFold Models) domain assigner and ECOD develop285 as a reference [28,30]. These proteomes included 16 notable model organisms such as *Danio rerio* and *Drosophila melanogaster*, as well as 32 organisms involved in diseases relevant to global health, such as *Plasmodium falciparum* and *Mycobacterium tuberculosis* (S1 Table). In total, we classified 746,349 domains from 536,808 proteins comprised of over 226,424,000 amino acid residues. By comparison, the reference version of ECOD (develop285) used for classification contained 898,379 domains from 599,852 peptide chains in 179,294 PDB depositions. These domains and their boundaries have been made available on the ECOD website (http://prodata.swmed.edu/ecod/index_af2.php) and have been deposited in a Zenodo repository [33].

DPAM classifies regions of AF2 models using sequence and structural aligners combined with the AF2 predicted aligned errors (PAE) to assist with determining globular regions, these regions are then assigned to ECOD using a neural network. In contrast to previous automated methods used to assign domains to ECOD, DPAM domains can be determined without necessarily containing a strong homologous link to the reference (see **Methods**). Per-residue domain classification categories for the 48 classified organisms reveal that eukaryotes contain more disordered and partially classified regions than either bacteria or (the single) archaea. The lower classification rate for domains in eukaryotes from AFDB predicted proteomes has previously been observed [31] and is likely due to some combination of prediction errors as well as contextual errors from regions that are only structured in the context of the correct higher-order multi-domain protein or protein complex. On average, DPAM returned classifications for 98% of the residues attempted (**Fig 1**). 66% of residues considered were confidently assigned as a domain to an ECOD H-group, whereas 4% were a globular domain but could not be confidently assigned to ECOD. 12% of residues classified by DPAM were determined to be flexible linkers or disordered regions, an additional 12% had overall low prediction confidence (<70 plDDT) and were not assigned as domains. 1% of residues were partial domains, which can be pseudogenes or genome annotation errors. Finally, 4% of residues had a simple topology, usually 2 or fewer secondary structure elements, and could not be assigned.

Two mycobacterial proteomes were anomalous compared to other bacteria. *M. ulcerans* (**Fig 1**, mycul) contained a higher fraction of its proteome assigned as partial domains (8%) and flexible (11%). *M. leprae* (**Fig 1**, mycle) contained a higher fraction of residues (16%) contained within globular domains that could not be assigned. Consequently, these proteomes had the lowest fraction of residues (among bacteria) with well-assigned domains by DPAM, 67% and 60% for *M. leprae* and *M. ulcerans*, respectively. *M. ulcerans*, the causative agent for Burulli ulcer, is believed to have recently diverged from environmental *M. marinum* through gene loss and horizontal gene transfer [35]. These underlying events may have led to the increased discovery of partial domains. *M. leprae* has one of the smallest proteomes (1602 proteins) among the set. Like many obligate pathogens, it has experienced genome reduction in comparison to other environmental species [36]. We suggest that this genome loss, and the consequent gain in pseudogenes, may explain the anomalous domain assignment behavior in this study. The malaria-causative eukaryote *P. falciparum*, (**Fig 1**, plaf7) also had the lowest fraction of residues determined to be in well-assigned domains, due to the high fraction of low prediction confidence regions in the underlying protein predictions. *P. falciparum* consequently had the lowest fraction of its proteome (41%) assigned to domains with well-established homologous links.

We also examined the population of structural class, ECOD architectures, and ECOD homologous groups among the collected well-assigned domains from the 48 proteomes. Population differences between taxonomical groups in domain classifications are well documented, and we were interested to what extent they held among these domains [15,37,38] (**Fig 2A**). We gathered ECOD architectures in five classes, similar to other structural domain classifications:

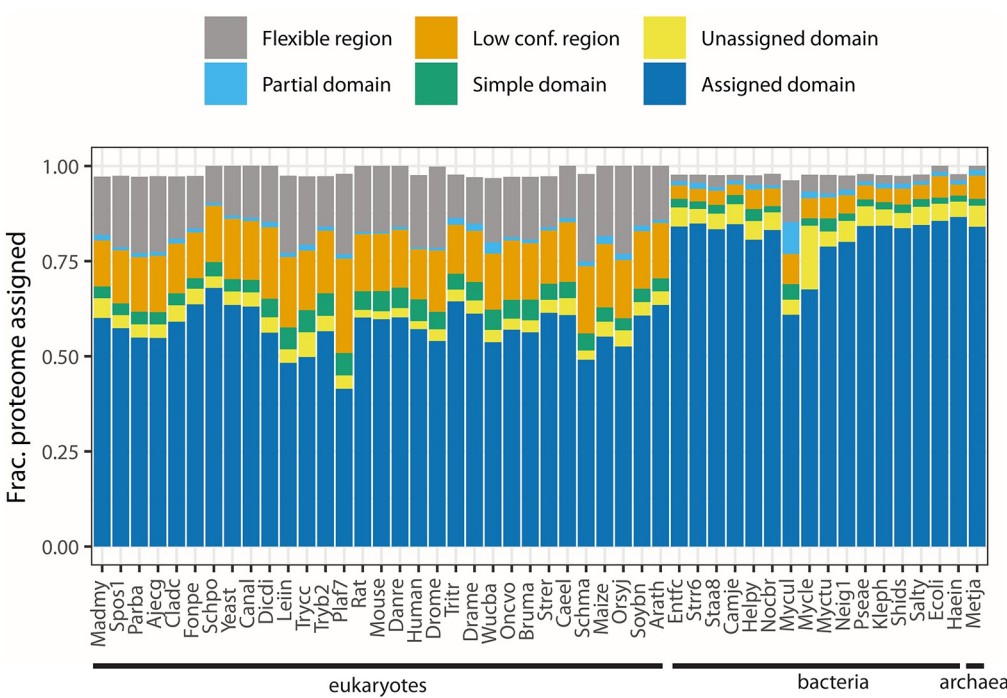

**Fig 1. Fractional coverage of proteomes by DPAM classification. DPAM classifies regions of AlphaFold2 models into one of six categories**: 1) Assigned domains (blue) are globular domains with well-defined homologous links to a reference domain 2) Unassigned domains (yellow) are globular with well-defined secondary structure but a confident automated link to a reference domain could not be determined. 3) Flexible linkers / disordered regions (gray) that are determined by PAE to be non-globular and in low-confidence interactions with other residues. 4) Partial domains (cyan) possess high-confidence links to reference domains where no high-coverage alignment can be found. 5) Simple domains (green) contain 2 or fewer secondary structure elements. 6) Low confidence regions (orange) are regions that cannot be assigned to domains and have low (<70 plDDT) confidence. Organisms are ordered by their NCBI taxonomy identifier from a tree built with the NCBI Taxonomy CommonTree tool such that organisms with the same domain, phylum, genus, etc. are grouped [34].

1) all-α proteins 2) all β proteins 3) α/β 4) α+β, and 5) a few secondary structure elements (SSE) and compared the population (by domain) of groups within these classes. The all-α class is the most populated (31%), followed by the all-β (21%) and α+β (20%). The all-α class is dominated by the α bundle architecture within which we find homologous groups of the Armadillo (ARM) repeats, Helix-turn-helix (HTH) domains, G-protein coupled receptors (GPCR), major facilitator superfamily (MFS) transporters, and Histone-like (Hist) domains present in the 20 most populated homologous groups (**Fig 2B**). The ARM repeat group is the most populated among the all-α architectures. This homologous group incorporates several short helical internal repeat proteins, among them the tetratricopeptide (CTPR) repeats. The uncharacterized CPTR repeat (UNP: A0A1X7YG12) is subdivided into multiple smaller sets of CTPR repeats (**Fig 2C**). In ECOD, like other domain classifications, small internal repeat proteins such as the CTPRs, leucine-rich repeats (LRR), and the HEAT repeats may be divided into smaller, more commonly observed, subunits [39]. This division of repeats into multiple domains may account for some of their enrichment among the classified domains. The helix-turn-helix (HTH), immunoglobulin-like (Igl), and β-β-α zinc fingers (BBA Zn) are also highly populated within our domain set. Disease-related protein 1 (DRP) from rice (UNP: Q84QL4), contains a "winged" HTH domain (**Fig 2D**, magenta). While both the LRR domain and the P-loop NTPase domains were previously classified by sequence, the HTH remained unannotated in sequence databases before structural prediction and domain classification. The DRP winged

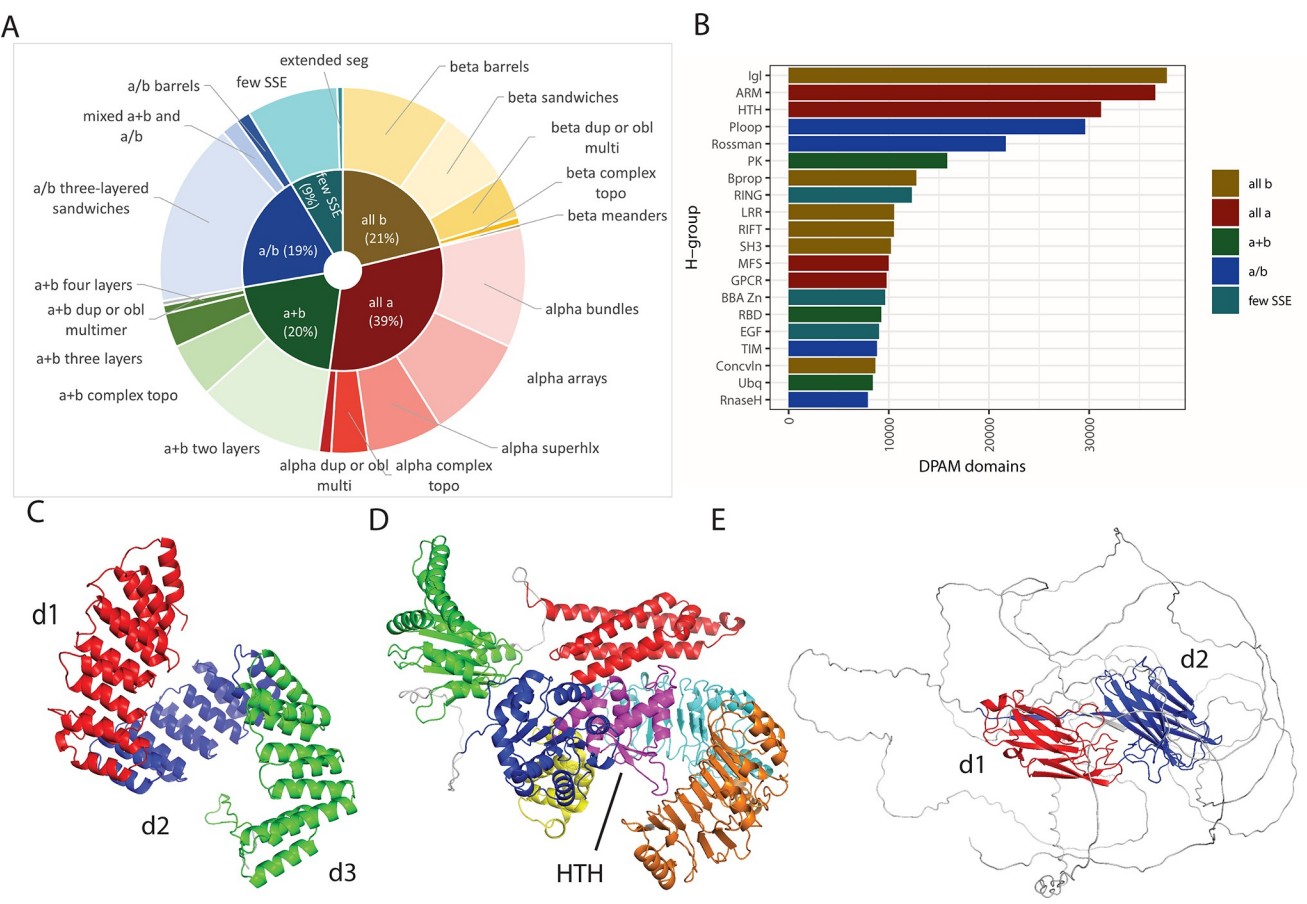

**Fig 2. Most populated ECOD groups of DPAM domains in AlphaFold DB whole proteomes.** A) Relative populations of DPAM domains assigned to ECOD by class (inner pie) and architecture (outer donut). B) Twenty most populated ECOD homologous groups from all AF2 whole proteomes, colored by class. C) An uncharacterized maize CTPR repeat (AFDB: A0A1X7YGI2). Internal repeat proteins (such as CTPR) can be subdivided into smaller repeated domains(d1:red, d2:blue, d3:green). D) Rice disease-related protein 1 (DRP1, AFDB: Q84QL4). The HTH domain of this protein (purple) is shown in the context of other detected domains. E) Fibronectin-binding protein B (FnbpB) of *S. aureus* (AFDB: Q2G1T5).

HTH assignment agrees with the CATH-Gene3D (G3DSA: 1.10.10.10) structural classification of this predicted protein [31,40]. Within the all-β proteins, the β-sandwich architectures are the most populated, among which the immunoglobulin domains (Igl) are the most populated homologous group. The Igl domains are well-known components of eukaryotic acquired immunity systems and are also commonly found as cell-recognition domains (**Fig 2E**) in surface proteins in bacteria. Fibronectin-binding protein B (FnBnpB) is an example of a prediction wherein much of the protein (formed of small repeats) is predicted with low confidence or disordered, but the Igl recognition domains are well-formed and possess a clear topological link to other β-sandwiches. FnbpB is implicated in pathogenesis and cell recognition. Two FnbpB IgL domains (red, blue) are modeled and detected amongst characteristic AF2 disordered regions. Domain classification of predictions can rescue regions of the protein that might escape recognition amid an overall low-confidence prediction.

Commonalities and differences between the classification of eukaryotes and bacteria

The whole proteomes in AFDB can be subdivided by taxonomy into eukaryotes, bacteria, and (a single) archaea (**S1 Table**). The eukaryotes include multiple plants (*O. sativa*, *Z. mays*, *A. thaliana*, *G. max*), mammals (*H. sapiens*, *R. norvegicus*, and *M. musculus*), and fungi (*S. cerevisiae*, *S. pombe*, *C. albicans*, among others). Numerous pathogenic yeast such as *A. capsulatus*,

*C. carrioni*, *F. pedrosi*, and *M. mycetomatis*) were released after the initial release as organisms of global health interest. Additionally, multiple helminth parasites such as *B. malayi*, *O volvulus*, *T. cruci*, *T. brucei*, *S. mansoni*, *S. schenkii*, and *S. stercolais* were released by AFDB and classified by us here. The bacteria in this set included the bacterial model organism *E. coli*, as well as multiple pathogenic bacteria such as *S. aureus*, *H. influenzae*, *L. infantum*, *M. tuberculosis* (and the aforementioned *M. ulcerans* and *M. leprae*), *N. gonorrhoeae*, *S. typhi*, *S. pneumoniae*, and *S. dysenterieae*. A single member of the archaea, *M. jannaschi* was also included as a model organism. This set includes five proteomes we previously published as a component of our human classification [27] using an earlier version of our domain partition and assignment algorithm (DPAM, see **Methods**) that has subsequently been improved. They are presented here in the context of comparison to other whole proteomes. We chose whole proteomes as a target for this analysis to understand the behavior of our algorithm on results unbiased by clustering and to observe potential orthologs and paralogs that might be removed by a larger clustering method. In the future, we anticipate that the results herein will guide our target selection for proteins from larger (and sparser) data sets.

We observed that a lower fraction of individual eukaryotic proteomes was well-assigned ($57.6 \pm 5.5\%$) on average than bacteria ($81.1 \pm 6.9\%$). Because of this difference, we more closely examined the well-assigned domains from those proteomes and enumerated the commonalities and differences among the most populated ECOD homologous groups (**Fig 3**). Among the eukaryotes, the ARM repeats (discussed above), the β-β-α Zinc fingers (BBA Zn), the Igl domains, and the HTH domains are the most populated homologous groups (**Fig 3A**). On the other hand, the most populated bacterial homologous groups are the HTH domains, the Rossmann-fold domains, the P-loop NTPase domains, and the TIM barrels (**Fig 3B**). Common among the most populated groups in both eukaryotes and bacteria, the Rossmann-folds, the P-loops, the Igl domains, the ARM repeats, and the HTH domains are the most populated groups. When the relative domain populations are stratified by species and normalized across species and homologous groups, we can more clearly see the contribution of certain species and H-groups to the domain population (**Fig 3C**). The BBA Zn fingers (**Fig 3D**) and the RING fingers (**Fig 3E**) (both types of Zn finger domains) are commonly found as components of eukaryotic transcription factors, often in repeats [41], and are more enriched in mammals and zebrafish. We found a similar enrichment of zinc-binding domains in our earlier classification of the predicted structures of the human proteome [27]. The EGF-like domain (**Fig 3F**), a small disulfide-bonded pair of β-sheets, is commonly an extracellular signaling factor but is implicated in a variety of roles in the extracellular environment, which likely explains their relative lack in bacteria and fungi. The most populated bacterial homologous groups are also found in eukaryotes at lower abundance. Rossmann, P-loop, and TIM barrel domains are ubiquitous homologous groups found across all domains of life encoding several enzymatic activities. The periplasmic-binding proteins (PBP-II) are commonly used by bacteria as environmental sensors for small molecules [42]. A principal difference between the eukaryotic and bacterial populations is that in eukaryotes the selective pressure against repetition and duplication is lower, and the differences between most populated eukaryotic and bacterial homologous domains reflect those homologous groups that are found repetitively (i.e., as multiple neighbors within the same protein) in eukaryotes such as the Igl domains, ARM repeats, and BBA and RING Zn fingers.

## Differences between the classification of experimental and predicted structures

The predicted AF structure domains were classified using a reference ECOD set determined from experimental structures. The respective populations of the most populated experimental

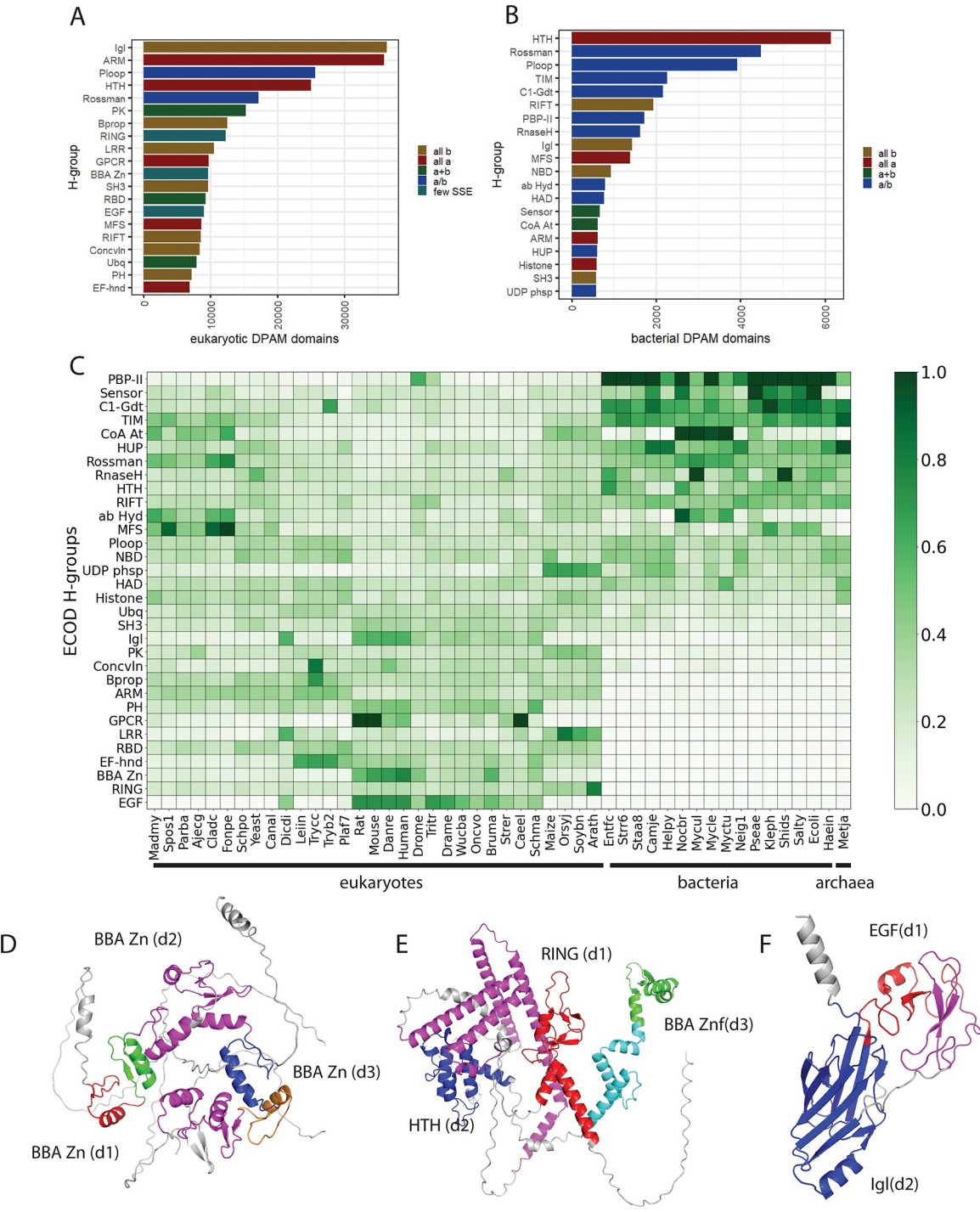

**Fig 3. Comparison of DPAM domain population in eukaryotes and bacteria from 48 AF2 DB proteomes.** A) Top 20 most populated homologous groups in domains classified from AFDB eukaryotes colored by structure class B) Top 20 most populated homologous groups in domains classified from AFDB bacteria colored by structure class C) Most populated homologous groups (>0.5% domain population) domain population stratified by species and normalized by total species and homologous group population(i.e. row and column).; D) *B. Malayi* C2H2-type domain-containing protein (AFDB: A8PDW8) contains three (d1:red, d2:blue, d3:green) repeated well-defined BBA zinc finger domains, as well as four ZnF repeats that were filtered for lacking sufficient topology (magenta). E) *C. elegans* RING finger-containing protein (AFDB: Q93343) contains one well-assigned RING finger ZnF domain (d1,red), along with HTH (d2, blue), and BBA Znf (d3, green) domains, also show filtered simple topology domains (magenta) along with a low-pLDDT region (cyan) F) C. elegans (AFDB: O45201) Delta-like protein dsl-1 contains one well-assigned EGF domain (d1, red) along with one immunoglobulin-like domain (d2, blue) as well as a EGF-domain with poorly formed topology (magenta).

groups have been discussed previously [15,43]. Domain population among experimental structures of proteins is not random, it is determined to an unknown degree by investigator interest and the ease of characterization. Structures difficult to characterize or unlikely to provide novel functional or structural information are less likely to be structurally determined. This likelihood may also vary over time as the structural determination expertise is gained overall and different research topics ascend or descend in the general interest. To examine the difference between our experimental reference set and the predicted set we were analyzing, we also calculated the population of homologous domains in our ECOD reference sets from different years: both our 2022 develop285 and our 2016 develop45 for comparison (**S1 Fig**).

The 2016 and 2023 experimental structure landscapes contain multiple structures of very similar or identical proteins (>99% sequence identity) (**S2 Fig**). Both sets were reduced in redundancy by sequence clustering (see **Methods**). The landscape of commonly used experimental methods has also been changing since the release of ECOD, so we also compared the balance of the most used structure determination methods for both experimental sets. Finally, using available metadata, we compared the distribution of eukaryotes and bacteria among the homologous groups in our experimental dataset to those DPAM domains from the structure predictions. Among the most populated ECOD homologous groups, the immunoglobulin-like (Igl) β sandwiches and the mixed α/β Rossmann folds remain the most populated ECOD homologous groups (**Fig 4A**). Igl domains are an important component of the human acquired immunity system and are also often found in bacterial cell-surface recognition proteins for pathogenesis. Rossmann folds are ubiquitous across organisms and are a common framework for numerous enzymatic reactions and binding activities [2]. Although the number of structures has dramatically increased in the intervening years (from 94,914 to 179,295) and the fraction of domains in ECOD coming from EM structures also has increased (from 1% to 9%), we did not observe a significant change in the relative ranks (by non-redundant F70 population) of the most populated homologous groups resulting from this change in methodological focus (**Fig 4B**). Similarly, although the number of domains from bacteria and eukaryotes increased significantly since 2016, the relative fraction of those domains in ECOD did not (46% and 40%, respectively) (**S3 Fig**).

We compared the relative population of the most populated eukaryotic and bacterial homologous groups in the AF structure predictions compared to their respective experimental references. Identification of groups with large relative expansion to their experimental reference may indicate groups 1) that may have been under-sampled by experimental methods 2) that may benefit disproportionately from inspection and analysis 3) that may have undergone biological duplication and expansion. 223 of 2485 ECOD homologous groups present in the AF2 eukaryotic domains differed significantly from the reference ($p < 0.05$, see Methods). Among the eukaryotic H-groups (**Fig 4C**) we noted the Igl domains (H: 11.1), ARM repeats (H: 109.4), and the major facilitator family (MFS, H: 5050.1) differing significantly from the eukaryotic reference. The Igl domains are slightly underrepresented in AF2 domains with respect to the reference, whereas the ARM repeats and MFS domains have increased representation in the DPAM domains. This is likely due to a combination of factors that lead to the structural determination of diverse Igl domains such as their relevance to human disease, and the subsequent number of known sequence families and sequence variants encountered among those families. Nearly 36% of the eukaryotic representative Igl domains in ECOD are human, whereas in the AF2 eukaryotes merely 11% of the eukaryotic domains are human. The increase in the representation of ARM repeats Is likely due to the sharp increase in the number of plant proteins characterized, resulting in 183,000 ARM domains found in *Arabidopsis*, rice, corn, and soybean. Similarly, the dramatic increase in the number of domains classified in the major facilitator superfamily (MFS) homologous group, is likely due to the difficulty in

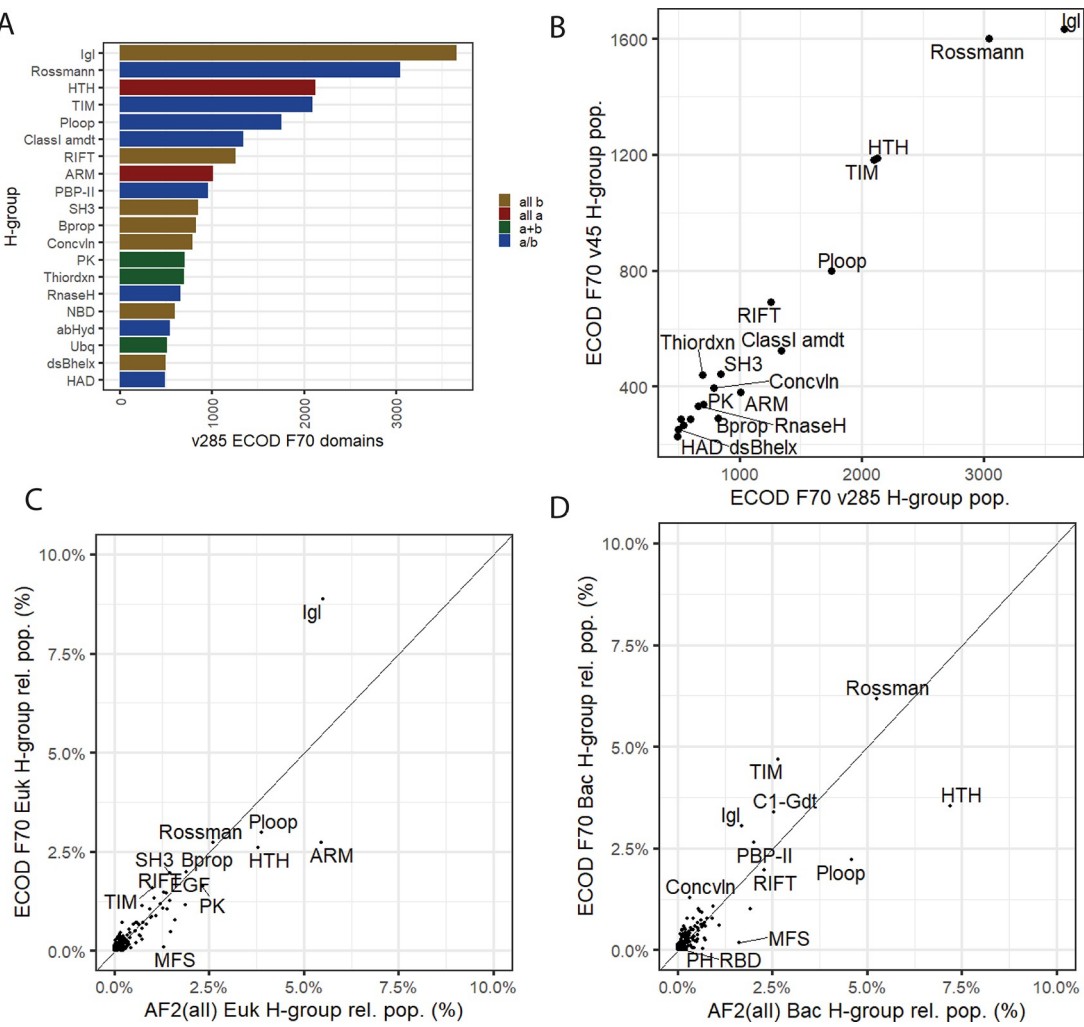

**Fig 4. DPAM domain classification compared to ECOD experimental reference.** A) Top 20 most populated homologous groups in ECOD v285 reference set, composed entirely of experimentally determined proteins and their domains B) Non-redundant (F70) domain population of ECOD v285 and v25 (i.e., 7 years earlier). C) Relative domain population of ECOD F70 eukaryotic representative domains compared to relative domain population of AF2 eukaryotic domains. D) Relative domain population of ECOD F70 bacterial representative domains compared to relative domain populations of AF2 bacterial domains.

structurally characterizing these well-known transmembrane domains. Among the bacterial H-groups, (**Fig 4D**) the HTH, P-loop, and MFS domains showed a significant increase with respect to the experimental reference, whereas Rossmann folds, TIM barrels, and Class 1 glutamine amidotransferase-like domains showed a slight but significant decrease in relative population in the predicted domains compared to the experimental reference. In conclusion, the relative domain population of homologous groups differs significantly between the "known" experimental protein domain world and the predicted whole proteomes offered by AFDB, even when controlled for the relative difference in domains of life between the two sets.

The transmembrane MFS domains transport a variety of small molecules across biological membranes. They have a complex α bundle topology and operate in pairs by a conformational shift. Perhaps, the most well-known MFS domain is lactose permease (**Fig 5A**), encoded by the lac operon [44,45]. ECOD classifies these domains as a singleton X-group (i.e., no known homologous groups with similar evolutionary history). Of the 82 ECOD F70 MFS domains,

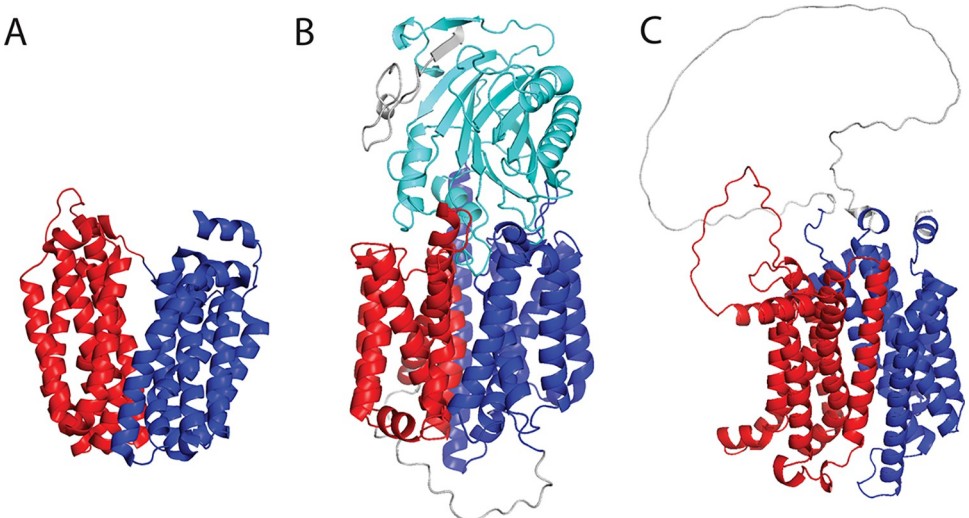

**Fig 5. Major facilitator superfamily domains in experimental and predicted structures.** A) *E. coli* lactose permease (PDB: 1PV7) and its two defined ECOD domains: e1pv7A1 (red), e1pv7A (blue).2). B) PGAP2-interacting protein (UNP: Q9H720) with two previously unannotated MFS domains (red, blue) and a known DNase I-like (cyan) domain C) solute carrier family 45 member 3 (UNP: Q96JT2) contains an MFS domain pair (red, blue) with a large insertion in the C-terminal domain. The annotation of this C-terminal domain is truncated in InterPro annotations.

the majority (67%) are from bacteria, while the remainder are eukaryotic (principally human, rat, and *Arabidopsis*). By comparison, 86% of the 9,981 MFS domains found in the AF2 structures predictions were found in eukaryotes. The MFS domains are well-studied, but difficult to structurally characterize, being both transmembrane and exhibiting a conformational shift between two states in transport. We examined the ECOD classification of these domains (wherein an MFS protein comprises two domains) and compared them to the well-studied sequence classifications (e.g., CDD [46], SMART [47], PROSITE [48], Pfam [9,40,49], SUPER-FAMILY [11]) indexed and collected by InterPro. Among the MFS domains from bacteria and eukaryotes classified here, 93.4% and 98.7% were covered by an existing sequence classification in InterPro. Some MFS domains are likely missed by sequence methods, but the principal reason a DPAM MFS domain was not covered by InterPro was that it originated from an AF2 deposition associated with an obsolete UniProt entry. We excluded predicted structures from an obsolete UniProt record for this analysis. Among the human MFS domains, we found only three domains detected by DPAM that were not annotated by the sequence classifications in InterPro. PGAP2-interacting protein (**Fig 5B**, UNP: Q9H720) consists of a pair of MFS domains with a C-terminal exonuclease domain (H:26.3) that is classified by InterPro (IPR036691). The structure of solute carrier family 45 member 3 (**Fig 5C**, UNP: Q96JT2) has an MFS domain pair but only the N-terminal domain is annotated by InterPro, the C-terminal domain contains a large disordered low-confidence insertion which likely prevents its detection by sequence methods. When domains from obsolete UniProt records were excluded, the fraction of domains previously annotated by sequence databases was 75.0% and 75.8% for eukaryotes and bacteria, respectively. This suggests that among larger well-understood protein families, we are seeing a convergence between sequence and structural classifications facilitated by highly accurate structure predictions. At the same time, among homologous groups that are less populated, structural domain classification is still finding domains previously unannotated by sequence domain classifications.

There were DPAM domains determined to be globular but with low confidence in their assignment. These unassigned domains may have previously unobserved topologies, domains

with known topologies whose homology is too distant to detect, or assignment errors arising from inconsistencies or boundary conditions in either DPAM or ECOD. In some cases, these domains could be candidates for seeding new homologous groups in ECOD (either H- or X-groups). We examined the length, secondary structure content, and sequence family assignments (by InterPro/Pfam where available) of sequence cluster representatives from this set of domains. Although a full manual curation of unassigned domains is the focus of future work, we discuss the properties of domains within this set and illustrate an example of a fast-evolving protein domain that is likely indicative of the types of curation necessary to classify this set.

The 72,918 low-confidence domains can be reduced to 38,260 representative domains by sequence similarity (see Methods). These unassigned representative domains have an average length of 99.8 ± 64.8 residues. By architecture, the largest fraction is all-α (46%), with a smaller fraction (36%) containing low or no detectable secondary structure. Some (11%) of the unassigned domains contained mixed α+β secondary structure, while the smallest fraction was the all-β domains (3%). All-alpha bundles are known to be difficult to assign, small changes in arrangements or lengths of helices in up-down bundles can make it difficult to establish homology by either sequence or structure. Domains with low secondary structure or where their evolutionary context is best understood through sequence or non-SSE mediated interactions (e.g., disulfide bonds or metal binding sites) such as the zinc finger transcription factors can be difficult to assign without a family-specific treatment.

We also considered whether the protein range of unassigned domains was coincident with known sequence classifications. Using InterPro as above, we compared the coverage of our unassigned domains to known InterPro domain classifications. Sequence families with strong conservation that can be identified can nevertheless have unknown structure and/or function (i.e. domains of unknown function or DUF families). Of the unassigned representatives, 15% have an overlapping domain annotation in InterPro, whereas 85% have not previously been annotated by sequence. Of the domains with a previous annotation. We examined the sequence-annotated unassigned domains for evidence of novel topology. In many cases, these domains are distantly homologous to existing homologous groups in ECOD. Here we present the classification of one of these cases into the yeast-killer toxin-like X-group as an example of how we would generally convert these putative domains into ECOD representatives.

The yeast-secreted protein CSS2 (UNP: P43600) belongs to Duf3676, which is limited to proteins from fungi and lacks an experimental structure representative. The CSS2 AF model adopts a circularly permuted α-β plait-like fold with an additional C-terminal edge strand packing against the sheet (Fig 6A) and was not confidently assigned to the ECOD hierarchy. The representative Duf3676 domain (UNP: P40049), a yeast ortholog, belongs to the uncharacterized protein YER076C. YER076C adopts the same core permuted alpha-beta plait topology, but the C-terminal edge strand extends the sheet facing the opposite direction as the CSS2 C-terminal strand (Fig 6B). The two yeast Duf3676 domains share a common disulfide formed between the loops following the N-terminal helix and the second strand of the core fold. The CSS2 protein includes an additional disulfide in a small N-terminal subdomain, while YER076C has two disulfides involving cysteine residues in the core fold. Another fungal Duf3676 domain (UNP: B8NBN6) has two of the three disulfides found in YER076C, but the AF model extends the sheet with a strand from the N-terminus (Fig 6C). Comparisons of the Duf3676 AF models highlight the common core elements of the fold that are decorated with variable regions from the N-terminus and the C-terminus of the protein.

While DPAM did not assign CSS2 to any ECOD homology group, its top identified parent domain (e6greb2, DPAM score 0.54) is a tandem duf26 ectodomain from the plant cysteine-rich receptor-like protein PDLP5. The duf26 ectodomain adopts the same permuted alpha-beta plait topology as the core fold of Duf3676, with the variable edge strand contributed from

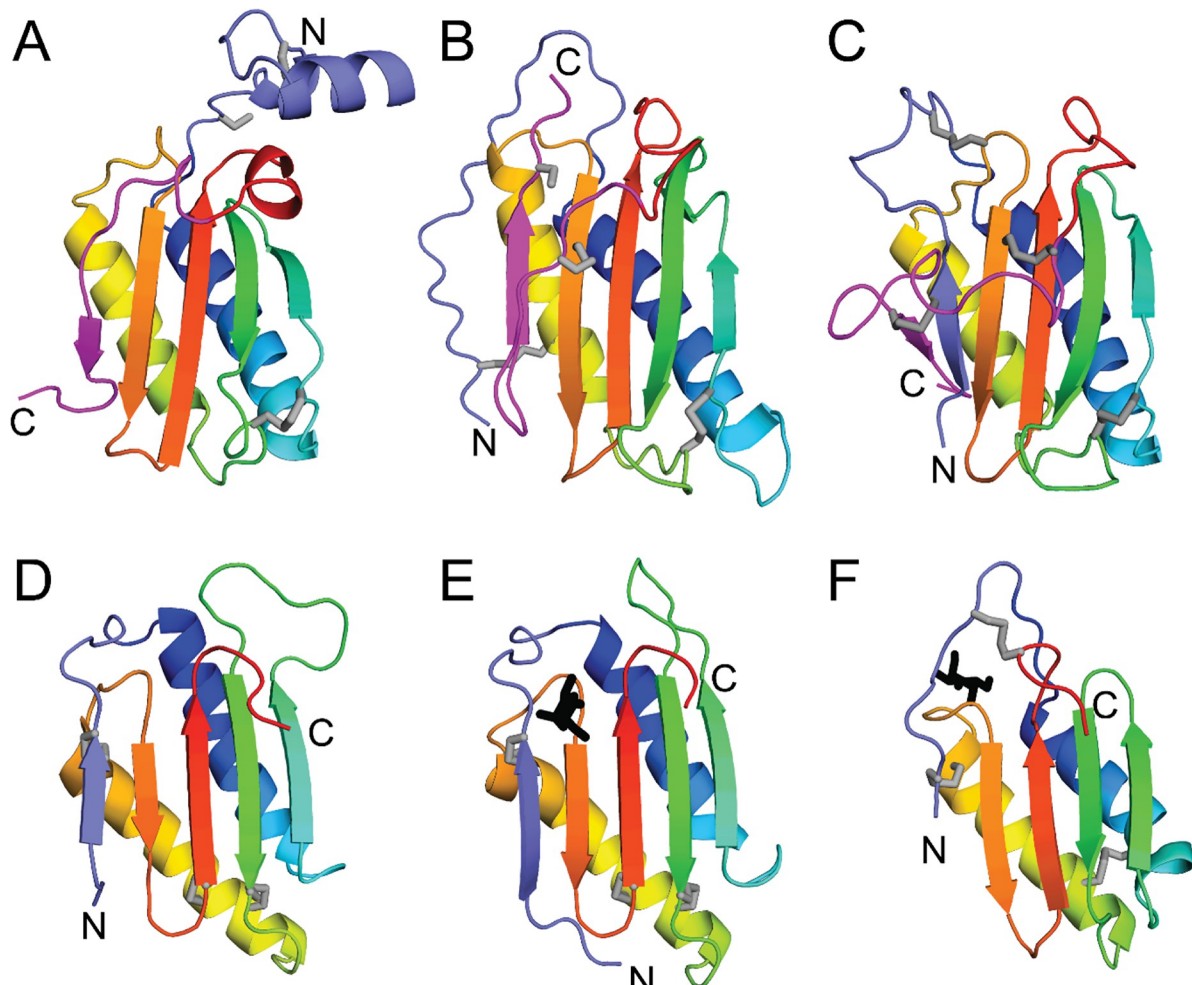

**Fig 6. Orphan domain Duf3676 represents a fast-evolving killer toxin-like fold.** Core fold common to Duf3676 (A-C) and yeast killer toxins (D-F) are depicted in rainbow cartoon from blue (N-terminus) to red (C-terminus). Cysteine side chains (gray stick) mark the disulfides, and termini are labeled. **A)** Duf3676 CSS2 AF model has an N-terminal extension (slate) to the common core fold that is decorated by an edge strand from the C-terminus (magenta). **B)** Duf3676 representative YER076C colored as in A, with the C-terminal strand in an opposite orientation. **C)** Duf3676 domain with an N-terminal strand (slate) replacing the variable edge strand. **D)** Top identified DPAM domain (ECOD: e6greB2) from Duf26 Gnk2 homolog has a similar N-terminal edge strand (slate) as Duf3676 domain in panel C, but alternate disulfide patterns (gray stick). **E)** Gnk2 binds carbohydrate (black stick) near the N-terminal strand. **F)** Yeast killer toxin bound to carbohydrate (black stick).

the N-terminus (Fig 6D). The PDLP5 Duf26 ectodomain belongs to the antifungal protein ginkbilobin-2 (Gnk2) homology group in ECOD, which includes a Gnk2 domain with the same disulfide patterns bound to a carbohydrate (Fig 6E). The Gnk2 homology group is classified together with the yeast killer toxin-like homology group. The yeast killer toxin group includes α-galactosyl binding *Lyophyllum decastes* lectin (LDL), which retains one disulfide found in Gnk2 but binds carbohydrate in an alternate position (Fig 6F). Despite these carbohydrate binding differences, both use the variable N-terminal decorations to the core fold to bind carbohydrate. The common topology found in the yeast killer toxins, combined with the taxonomic distribution of Duf3676 among fungi, suggest the two are related. The lower confidence DPAM assignment of Duf3676 to yeast killer toxins reflects fast-evolving folds marked by alternate disulfide patterns and variable decorations to the core that likely arose to diversify

function. The assignment suggests the Duf3673 domains might also function as lectins that bind carbohydrate through the variable regions of the fold.

The domain classification of these proteomes was performed to determine the balance of homologous groups in different organisms, to develop techniques for broadly filtering low-confidence and poorly assigned domains from the final sets, and to establish the performance of both AlphaFold and DPAM on different domains of life. Future domain classification by DPAM of larger data sets will necessarily be on clustered data where full coverage of a given species will not be guaranteed. By performing these classifications, making them publicly available, and showing the convergence of our classification with other types of classifications, we have shown that predicted structures can be classified in a high-throughput method and that a hybrid domain classification of experimental and computationally predicted structures can add to the broader base of scientific knowledge. With even larger data sets ahead of us in the future, we consider broader representative sets encompassing the entire known protein universe, or classifications from multiple versions of the same computational method, or different methods altogether, we will need to tighten our criteria for selection to include only proteins representative of large clusters of unclassified proteins. This whole proteome classification will serve as the basis of these future clustering efforts and our larger classification sets.

## Methods

### Domain classification of AlphaFold-predicted structures using DPAM

Predicted structures and PAEs of 16 "model organism" and 32 "global health" (S1 Table) proteomes were downloaded from the AlphaFold2 database [22]. These structures were classified into domains using the most recent version of the DPAM classification pipeline, previous versions of which have been used to classify models of both *Vibrio parahaemolyticus* [50] and *Homo sapiens* [27]. The principal improvements of the current version (v5) over previous versions are: 1) a modification to the routine that merges short, over-split, regions into single domains and 2) a more explicit categorization of types of regions and domains recognized. 490,060 proteins (~212M residues) were classified using DPAM v5. Human proteins predicted as overlapping models were specifically excluded from this analysis (3095 models from 1161 proteins). We note that at the time of writing, one reference proteome (*Trichuris tritis*: UP000030665), and 24,859 protein records were no longer extant on UniProt. Where possible, we determined superseding UniProt records by 100% sequence identity match (and identical length) between an obsolete record and an existing record. This mapping was used to extract metadata such as annotated sequence domains, functions, and names, but did not affect the overall domain classification, which was performed on AlphaFold Database depositions.

Using a combination of sequence [51] and structural aligners [52], as well as consideration of residue-wise segment. The details of the DPAM algorithm are described elsewhere [53], here we briefly describe the overall method as well as changes made since the initial implementation. Using the predicted aligned errors (PAE) distributed with the predictions, regions that appear disordered or as linkers are excluded. Several inter-residue measures, such as inter-residue distance, PAE, and shared membership within HHsearch and DALI hits in an ECOD reference set are converted into probabilities for two residues to be within the same domain based on regression analyses on the benchmark ECOD domains. These probabilities were then used to cluster 5 residue segments into domains.

DPAM classifies protein regions into four categories based on their physiochemical properties and their evolutionary signal. 1) "Well-assigned domains" are those globular domains with a strong similarity by structure or sequence to a known experimental ECOD reference. 2) "Unassigned domains" are those domains that appear globular by their PAE description but

lack a strong homologous hit within the ECOD reference. 3) "Partial domains" are domains with a strong evolutionary signal to a reference domain, but where the coverage to that domain is insufficient to meet our coverage threshold, these domains can often be pseudo-domains or the product of incorrect genome annotation. 4) "Simple topology" are those domains that lack sufficient secondary structure elements to be considered globular. These are often single helix or coiled-coil regions that are difficult to classify against ECOD. For this work, we also examined those regions that could neither be classified as globular domains nor assigned to the ECOD hierarchy and separated them into regions with low (plDDT < 0.7) prediction confidence (i.e. low confidence regions) and those with high prediction confidence (i.e. "flexible regions"). Protein structure figures were composed using PyMOL [54]. Plots and statistical analysis were conducted using R (ggplot2) [55,56] and Excel.

## Comparison against sequence classifications using InterPro

InterPro 96.0 data were downloaded in bulk from the InterPro website using the 'protein2ipr. dat'file. UniProt XML depositions were retrieved via individual REST queries for proteins of interest and then uploaded to a local PostgreSQL database for analysis.

## Selection of sequence representatives from families using CD-HIT

Domains from ECOD PDB are clustered by F-groups using CD-HIT sequence clustering [57]. Domains from those clusters are then selected by automated criteria. These representatives are generated at 40%, 70%, and 99% levels of sequence redundancy and offered as distributable files on the ECOD website (F40, F70, F99). These representatives are generated such that there is a minimum of one domain per F-group, but considerable sequence redundancy can occur between F-groups. A smaller number of domain representatives would be expected if you clustered over the set as a whole. We generated the sequence clusters using local alignments (-G 0) with coverage cutoffs of (0.9, 0.7, 0.7), sequence identity thresholds of (0.99, 0.70, 0.40), and word sizes of (5 4, 2) for F99, F70, and F40 representative sets respectively. For the selection of domain representatives, Previous representatives (i.e., determined in a previous version) are preferred, followed by manually curated representatives, and provisional manual representatives (i.e., automatically assigned "manual" representatives necessary to populate a newly used sequence family). If no such representatives are available, new domains from x-ray structures are preferred, with preference given to higher resolution and more recently released structures. For ECOD v285, publicly available representative sets were used. As ECOD v45 preceded our development of this method for selecting and generating representatives, they were newly generated here. They are made available in our data-sharing repository [33] and on our website.

## Selection of sequence representatives from unassigned domains using mmSeqs

Unassigned domains from ECOD AF2 were clustered into sequence representatives using the fast search program mmSEQs [58]. Domain FASTA sequences were generated from putative domain ranges and AFDB protein CIF files. Clusters were generated using 'easy-cluster' method and a bidirectional 70% coverage cutoff (-c 0.7 –cov-mode = 0). Representatives were selected by mmseqs.

## Significance analysis of relative population changes

Chi-squared analysis of H-group populations was conducted using the CHISQ.TEST function in Microsoft Excel. The significance level was set at $p < 0.05$, but the Bonferroni correction for multiple testing was applied based on the number of H-groups considered per set.

## Supporting information

**S1 Table. 48 Proteomes with DOL, identifiers, ordering, and domain classification status.**
(DOCX)

**S1 Fig. Total and representative domains in ECOD from 2016 (v45) and 2022(v285) by experimental method.** The increase in domains from the most used methods in the PDB. The increase in structures determined by electron microscopy does lead to a sizeable increase in (redundant) total domains as well as representative domain structures.
(DOCX)

**S2 Fig. PDB depositions and chains classified by ECOD in 2016(v45) and 2022(v285).** The methods with the most sizeable increase between 2016 and 2022 were X-ray crystallography and electron microscopy. Electron microscopy structures commonly have many proteins (chains) as reflected in this data.
(DOCX)

**S3 Fig. Domains of Life recorded in ECOD 2016(v45) and 2022(v285).** Domains of life determined by PDB metadata associated with NCBI Taxonomy Ids. Archaea (A), Eukarya (E), Bacteria (B), and Virus (V) along with Other (O) or Unknown (U) NCBI taxonomy categories. PDB metadata associated with obsolete PDB or NCBI taxonomy ids resolve to (Unk).
(DOCX)

## Acknowledgments

We acknowledge the Texas Advanced Computing Center (TACC) at The University of Texas at Austin for providing HPC resources.

## Author Contributions

**Conceptualization:** R. Dustin Schaeffer, Jing Zhang, Lisa N. Kinch, Qian Cong, Nick V. Grishin.

**Data curation:** R. Dustin Schaeffer.

**Formal analysis:** R. Dustin Schaeffer, Lisa N. Kinch, Qian Cong.

**Funding acquisition:** R. Dustin Schaeffer, Qian Cong.

**Investigation:** R. Dustin Schaeffer, Jing Zhang, Lisa N. Kinch, Qian Cong.

**Methodology:** R. Dustin Schaeffer, Jing Zhang, Kirill E. Medvedev, Qian Cong.

**Project administration:** R. Dustin Schaeffer, Nick V. Grishin.

**Resources:** R. Dustin Schaeffer, Nick V. Grishin.

**Software:** R. Dustin Schaeffer, Jing Zhang, Kirill E. Medvedev, Qian Cong.

**Supervision:** R. Dustin Schaeffer, Qian Cong, Nick V. Grishin.

**Visualization:** R. Dustin Schaeffer, Lisa N. Kinch, Qian Cong.

**Writing – original draft:** R. Dustin Schaeffer.

**Writing – review & editing:** R. Dustin Schaeffer, Jing Zhang, Kirill E. Medvedev, Lisa N. Kinch, Qian Cong, Nick V. Grishin.

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
