## [Decision Letter · Decision Letter 0]

28 Nov 2023

Dear Dr. Schaeffer,

Thank you very much for submitting your manuscript "ECOD domain classification of 48 whole proteomes from AlphaFold Structure Database using DPAM" for consideration at PLOS Computational Biology.

As with all papers reviewed by the journal, your manuscript was reviewed by members of the editorial board and by several independent reviewers. In light of the reviews (below this email), we would like to invite the resubmission of a significantly-revised version that takes into account the reviewers' comments.

Two reviewers mention availability of the data on the ECOD website. I think this is a good idea. I find one aspect of using ECOD that makes it difficult: finding what 3 or 4 number code describes an ECOD domain. For example, protein kinases are ECOD 206.1.1. This designation is used in the DPAM download files associated with this paper. If you look up a PDB code on the website, you do not get this information at all. There is nowhere to find that information on the ECOD website for any PDB entry, unless you download ecod.latest.domains.txt and grep for a PDB code. Sequence search does not currently work (I get "Job 1ZkGRk has errors").

We cannot make any decision about publication until we have seen the revised manuscript and your response to the reviewers' comments. Your revised manuscript is also likely to be sent to reviewers for further evaluation.

Sincerely,

Roland L. Dunbrack Jr., Ph.D.

Academic Editor

PLOS Computational Biology

Nir Ben-Tal

Section Editor

PLOS Computational Biology

Reviewer's Responses to Questions

**Comments to the Authors:**

Reviewer #1: In this work, Schaeffer et al. describe their efforts on the classification of known ECOD domains onto the proteomes of 48 species across the tree of life, now possible due to the availability of structural models in the AlphaFold database. This is a timely work, as it allows us to infer about the distribution of known domains and folds in Nature, which is not accurately depicted by the Protein Databank. The paper is well written, and the ideas are well conveyed, but I have some major and minor issues. Please see below:

Major:

- I am disappointed to not see a discussion on novelties or global but non-assigned predicted domains. The authors write that 66% of residues could be assigned a domain, whereas 4% couldn't but don't discuss these further. What are their properties? are they mostly repetitive, helical, stranded, small, large, etc? Could they give origin to new X-groups, for example?

- the authors mention in the abstract that they demonstrate that eukaryotic proteins are more repeated than bacterial ones, but that is not obvious from the main text. A figure highlighting the different tendencies for domain compositions between bacteria and eukaryota would allow to a better understanding of the different tendencies between the two different groups and would make the discussion stronger.

- how are these domain annotations found in ecod? it is not clear from the manuscript neither the ECOD webpage. I would expect that querying by uniprot ac would allow for finding the domains assigned to them, for example, but that does not seem the case.

Minor:

- figure 1: the xticks are not easy to read because they are cramped into each other and also are not full species names

- figure 2 legend: I think the labels of the figures in the legend text is shifted. B) appears twice. Also, it says "The HTH domain of this protein (purple) is shown in the context of other detected domains" but this is difficult to grasp in the figure. the three structure panels would be easier to interpret if the different domains were labelled. same in figure 3.

- figure 3: in panel D, it is not obvious the separation between bacteria and eukaryotes. highlighting them as in figure 1 would facilitate the interpretation of the figure.

- p 8 line 134: Fig. 1 mycle, you mean "myucl"?

- p19 line 390: the citations are wrongly placed and misleading. they are for pymol, R and ggplot and not for excel.

- p 19, line 408: the authors write "lower resolution" but i guess what they mean is "higher resolution", ie. lower value

Reviewer #2: The manuscript “ECOD domain classification of 48 whole proteomes from AlphaFold Structure Database using DPAM” by RD Schaeffer et al (PCOMPBIOL-D-23-01629) describes a prospective extension of the evolutionary classification of protein domains (info on experimental structures in the ECOD database (http://prodata.swmed.edu/ecod/) towards models of individual protein structures produced by AlphaFold2 which are held in the AlphaFold Protein Structure Database (https://alphafold.ebi.ac.uk/). The results described in the manuscript comprise models from 48 proteomes (excluding human described in the previous work from the same lab). The authors found that ~75% proteins in the considered proteomes have high-confidence domains similar/homoogous to the domains previously identified for the experimental structures in the earlier ECOD release, maintained by the same lab. Additionally, ~ 5% (from Fig 1 in the manuscript) of proteins have domains that do not have any similarity/homology detected by the methods used. Authors also provided comparative analysis of new domains as well as of the annotated domains in different kingdoms of life. In general, subject of the manuscript is extremely relevant for the field of structural molecular biology and worth publishing in PLoS Computational Biology. Below are some comments/suggestions that can, in my opinion, improve quality of the manuscript.

1. Title of the manuscript should reflect the fact that DPAM used in this study differ from earlier DPA used to annotate models of human proteome.

2. Number of domains provided on line 106 (p. 6 of the manuscript), 898 379, contradicts number 746 349 given in the abstract. The latter corresponds to the number of entries in the file af2_all_dpam_v5_domains.txt, available at Zenodo repository. Also, those numbers are surprisingly close to the numbers from previously released domain annotation for the experimental structures (line 108 on p.6). Authors should clarify and comment on that situation.

3. The data are actually not available from the main ECOD webpage and thus authors either should remove the mentioning of ECOD on line 109, p.7, or add corresponding links to the ECOD webpage. Also, it would be highly desirable if ECOD users could search for specific domains in both experimental and model AF2 structures.

4. In figure 1, color palette used in the figure disagree with the description of colors in the figure caption. For instance, assigned domains are in blue in the figure, but claimed to be gray in the caption. This should be corrected.

5. Ordering by taxonomy ID mentioned on lines 132 and 133 on p.8 should be explained more in details.

6. Short name ‘Mycle’ on line 134 should be actually ‘Mycul’.

7. Percentages given on line 155, p.8 are misleading as they refer to the percentage inside only one fragment of inner pie. That should be explained better. Additionally, Fig 2A would convey the information better if fractions/percentages will be added directly to the chart with explanation that fraction in the inner pie are with respect to the entire circle and fractions in the outer pie corresponds to one of the segments in the inner pie. Fig 2D would also benefit, if all non-magenta domains will be blurred. Also, lines 182 – 186 and 188 – 190 from the caption of this figure belong to the discussion in the main text rather then to the figure caption, which should, in my opinion, contain only factual info related to the displayed picture(s).

8. Panel C in fig 3 is redundant as all info is in panels A and B. Also, Reference to panel D is missing in the main text.

9. Fig 4B does not display relative population (as it is claimed in the main text and in the caption), thus it is difficult to judge whether it supports statement in the text or not. Also, sentence on line 279, p 14, belongs to the main text rather than figure caption.

10. Line 286, p.14. Should it be “under-sampled by experimental methods”?

11. Does “low confidence domains” refer to low conf region or unassigned domain in Fig 1?

Reviewer #3: The author used ECOD domain classification to annotate 48 proteomes from AlphaFold Structure Database. Overall, the paper has some significance, but there are still some important issues that need to be addressed.

1. The author concludes that eukaryotes contain a higher proportion of disordered regions and unassigned domains compared to bacteria. However, these findings seem to have been previously proposed, as referenced in the papers by Bordin et al. [1] and Porta-Pardo et al [2]. Specifically, Bordin et al. conducted a structural analysis of 21 protein complexes predicted by AlphaFold, and the difference appears to lie in the use of different annotation methods (CATH). Nevertheless, both studies reached similar conclusions. Therefore, I believe the author should emphasize their own strengths in light of these existing findings.

reference:

[1] Bordin, Nicola, et al. "AlphaFold2 reveals commonalities and novelties in protein structure space for 21 model organisms." Communications Biology 6.1 (2023): 160.

[2] Porta-Pardo, Eduard, et al. "The structural coverage of the human proteome before and after AlphaFold." PLOS Computational Biology 18.1 (2022): e1009818.

2. Did the author consider the partial domains as discontinuous domains? Although they may be discontinuous in sequence, they are close in structural space.

3. In order to identify inherently disordered regions, tools like IUPred can be used. Relying solely on the evaluation metrics of AlphaFold is limited when it comes to recognizing disordered regions.

4. The references of the paper need to be carefully checked. For example, reference 20 should be 23, the reference of the alphafold on page seven, and the position of references 52-54.

5. Authors should also carefully check that abbreviations are consistent throughout the paper. For example, AFDB and AF2DB.

**Have the authors made all data and (if applicable) computational code underlying the findings in their manuscript fully available?**

Reviewer #1: None

Reviewer #2: Yes

Reviewer #3: None

PLOS authors have the option to publish the peer review history of their article (what does this mean?). If published, this will include your full peer review and any attached files.

Reviewer #1: **Yes: **Joana Pereira

Reviewer #2: No

Reviewer #3: No
---

## [Decision Letter · Decision Letter 1]

20 Feb 2024

Dear Dr. Schaeffer,

We are pleased to inform you that your manuscript 'ECOD domain classification of 48 whole proteomes from AlphaFold Structure Database using DPAM2' has been provisionally accepted for publication in PLOS Computational Biology.

Best regards,

Roland L. Dunbrack Jr., Ph.D.

Academic Editor

PLOS Computational Biology

Nir Ben-Tal

Section Editor

PLOS Computational Biology

Reviewer's Responses to Questions

**Comments to the Authors:**

Reviewer #1: The authors addressed all my comments and the paper is now much improved. The data is now really easy to access in ECOD and the discussion of the novelties is sufficient and clear.

I have no more issues, just a very minor comment: in figure S3, "unknown" is in the plot twice ('U' and 'Unk').

**Have the authors made all data and (if applicable) computational code underlying the findings in their manuscript fully available?**

Reviewer #1: Yes

PLOS authors have the option to publish the peer review history of their article (what does this mean?). If published, this will include your full peer review and any attached files.

Reviewer #1: No

---

## [Editor Report · Acceptance letter]

23 Feb 2024

PCOMPBIOL-D-23-01629R1 

ECOD domain classification of 48 whole proteomes from AlphaFold Structure Database using DPAM2

Dear Dr Schaeffer,

I am pleased to inform you that your manuscript has been formally accepted for publication in PLOS Computational Biology. Your manuscript is now with our production department and you will be notified of the publication date in due course.

With kind regards,

Zsofia Freund
